# Intestinal Dominance by *Serratia marcescens* and *Serratia ureilytica* among Neonates in the Setting of an Outbreak

**DOI:** 10.3390/microorganisms9112271

**Published:** 2021-10-31

**Authors:** Elias Dahdouh, Fernando Lázaro-Perona, Guillermo Ruiz-Carrascoso, Laura Sánchez García, Miguel Saenz de Pipaón, Jesús Mingorance

**Affiliations:** 1Servicio de Microbiología, Hospital Universitario La Paz, IdiPAZ, Paseo de la Castellana, 261, 28046 Madrid, Spain; fernandolazaroperona@gmail.com (F.L.-P.); guillermo.ruiz@salud.madrid.org (G.R.-C.); jesus.mingorance@salud.madrid.org (J.M.); 2Servicio de Neonatología, Hospital Universitario La Paz, 28046 Madrid, Spain; laurasg_alcobendas@yahoo.es (L.S.G.); miguel.saenz@salud.madrid.org (M.S.d.P.)

**Keywords:** *Serratia marcescens*, *Serratia ureilytica*, neonatal ward, intestinal dominance, qPCR

## Abstract

(1) Background: We determined the relevance of intestinal dominance by *Serratia* spp. during a neonatal outbreak over 13 weeks. (2) Methods: Rectal swabs (n = 110) were obtained from 42 neonates. *Serratia* spp. was cultured from swabs obtained from 13 neonates (Group 1), while the other 29 neonates were culture-negative (Group 2). Total DNA was extracted from rectal swabs, and quantitative PCRs (qPCRs) using *Serratia*- and *16SrRNA*-gene-specific primers were performed. relative intestinal loads (RLs) were determined using ΔΔC_t_. Clonality was investigated by random amplified polymorphic DNA analysis and whole-genome sequencing. (3) Results: The outbreak was caused by *Serratia marcescens* during the first eight weeks and *Serratia ureilytica* during the remaining five weeks. *Serratia* spp. were detected by qPCR in all Group 1 neonates and eleven Group 2 neonates. RLs of *Serratia* spp. were higher in Group 1 as compared to Group 2 (6.31% vs. 0.09%, *p* < 0.05) and in the first swab compared to the last (26.9% vs. 4.37%, *p* < 0.05). Nine neonates had extraintestinal detection of *Serratia* spp.; eight of them were infected. RLs of the patients with extraintestinal spread were higher than the rest (2.79% vs. 0.29%, *p* < 0.05). (4) Conclusions: Intestinal dominance by *Serratia* spp. plays a role in outbreaks and extraintestinal spread.

## 1. Introduction

*Serratia* species have had a long history of causing nosocomial outbreaks. This is mainly due to their ability to persist for long periods of time in water environments and having the potential to cause a full spectrum of clinical infections [1]. These opportunistic pathogens, and especially *Serratia marcescens*, are a common cause of outbreaks in paediatric units, with significant rates of morbidity and mortality in immunocompromised patients and preterm infants [2,3]. *Serratia* infections may be particularly challenging due to their intrinsic non-susceptibilities to antimicrobial agents that include second-generation cephalosporins and tetracyclines [4]. Moreover, they have the potential to acquire resistances to additional antimicrobial agents, including the broad spectrum carbapenems [5]. This is particularly concerning since the source reservoir for infections is usually hard to determine [6].

Surveillance of intestinal carriage of multidrug-resistant organisms (MDROs) and other organisms of interest through rectal swabs is a widely accepted infection-control measure [7]. Though informative, one of its limitations is the lack of differentiation between the extents to which different patients are colonized. In this work, our objective is to go beyond the detection of intestinal colonization by *Serratia* spp. and retrospectively investigate the role of the relative intestinal loads (RLs) in the setting of an outbreak that lasted for 13 weeks. We demonstrate a link between intestinal dominance by *Serratia* spp. and the ability to escape the intestinal niche and cause outbreaks and infections.

## 2. Materials and Methods

### 2.1. Sample Collection from Neonates during the Thirteen Weeks of the Outbreak

This study is a retrospective analysis of an outbreak that was declared by the Preventive Medicine Department at the Hospital Universitario La Paz (HULP). The rectal swabs that were routinely collected for epidemiological surveillance at HULP during the period of the outbreak (from late December 2018 to early March 2019) were reused in this study. These swabs were obtained during a period of 13 weeks from 42 neonates admitted to the neonatal ward. During this time period, the outbreak declared by the Preventive Medicine Department was believed to be caused by *Serratia marcescens* and to affect 13 neonates. All the swabs obtained from these patients throughout the 13 weeks were collected. Additionally, rectal swabs obtained from 29 neonates from the same unit but not having any *Serratia* spp.-positive cultures were collected from week eight to week thirteen of the outbreak. After being used for routine screening, the rectal swabs were exhausted in 500 µL of TE buffer (10 mM Tris and 1 mM EDTA; pH = 8.0) and stored at −20 °C until used. In the samples where the routine culture was positive, ten colonies from each agar plate where the isolate was first detected for each patient were selected for storage in conservation cryogenic tubes at −20 °C until used.

### 2.2. Water Sample Collection

Four samples collected during week six by swabbing and collecting water samples from the pipes, basin, and sink drain at the neonatal ward were also included in the study. Samples from hard surfaces were obtained with a cotton swab moistened with a sterile saline solution. The swabs were then inoculated in brain-heart infusion broth (Tec-Laim, Madrid, Spain), vortexed for 30 s, incubated overnight at 37 °C, and plated on MacConkey agar (Biomérieux, Marcy L’etoile, France). Water samples were obtained by first extracting water using a sterile urinary catheter and syringe. The water was then centrifuged at 3000 rpm for 5 min, and the precipitate was inoculated in brain-heart infusion broth, incubated overnight at 37 °C, and plated on MacConkey agar (Biomérieux, Marcy L’etoile, France).

### 2.3. DNA Extraction

Total DNA was extracted by first mixing 100 µL of the suspended rectal swabs with 900 µL TE buffer in order to avoid quantitative PCR (qPCR) inhibition. The mixture was heated to 95 °C for 20 min, followed by mechanical lysis at 7000 rpm for 70 s using a MagNa Lyser (Roche^®^, Mannheim, Germany). DNA was then extracted using the MagNA Pure Compact system (Roche^®^, Mannheim, Germany) and stored at −20 °C until used.

### 2.4. Relative Quantification of the Intestinal Loads of Serratia Species

The relative quantification of the intestinal loads of *Serratia* spp. was performed through qPCR targeting the bacterial *16SrRNA* gene and a region of the *luxS* gene specific to *Serratia* spp. The primers used were 5′-TGGAGCATGTGGTTTAATTCGA-3′ and 5′-TGCGGGACTTAACCCAACA-3′ for *16SrRNA* [8] and 5′-TGCCTGGAAAGCGGCGATGG-3′ and 5′-CGCCAGCTCGTCGTTGTGGT-3′ for *Serratia* spp. [9]. These PCR reactions were performed in separate tubes. The reaction mixture included 10 µL PowerUP™ SYBR^®^ Green Master Mix (Applied Biosystems, Waltham MA, USA), 6 µL H_2_O, 0.05 µM of each primer pair, and 2 µL of template DNA. The qPCR reactions were carried out in a CFX Connect™ Real-Time System (BioRad, Madrid, Spain) with the following conditions: three minutes at 95 °C, followed by 40 cycles of 15 s at 95 °C, and 1 min at 60 °C. Melting-curve analyses were performed after each reaction. The threshold cycle (C_t_) values for the *Serratia*-specific gene were normalized to the C_t_s of *16SrRNA* (marker for total bacterial mass). These values were then normalized to pure cultures of two *Serratia* isolates involved in the outbreak using the ∆∆C_t_ method [10]. The results were expressed as relative intestinal loads (RLs) on an inverse logarithmic scale where a value of 0 represents 100% of the bacterial population, −1 represents 10% of the bacterial population, −2 represents 1% of the bacterial population, and so on until −6 (detection limit), which represents 0.0001% of the bacterial population.

To determine the efficiency and detection limit of the qPCR reactions, two *Serratia* isolates involved in the outbreak were cultured overnight on blood agar plates (Biomerieux, Marcy L’etoile, France). A 0.5 McFarland suspension was then prepared and serially diluted in 10-fold dilutions. qPCR was performed for each dilution, and the efficiency of the reaction was calculated as 10^−1/slope^ − 1 [11]. To test for the effect of dilution on the normalization, the ΔC_t_ values obtained were plotted against the different dilutions prepared. These experiments were performed in triplicate and showed that sample dilution did not affect the ΔC_t_ value (slope ≈ 0). The detection limit was 1000 CFU/mL, and the reaction efficiency was 94.4% (Appendix A).

### 2.5. Clonality Analysis

Clonality of the isolates obtained from the first culture-positive rectal swab of each neonate involved in the outbreak, in addition to an isolate recovered from the blood of one of the patients and the four environmental samples, was investigated by random amplified polymorphic DNA (RAPD) analysis. The primers used were OPA-2 (5′-TGCCGAGCTG-3′), OPA-12 (5′-TCGGCGATAG-3′), and OPA-18 (5′-AGGTGACCGT-3′). The PCR conditions were as previously described [12]. The profiles obtained after gel electrophoresis were analysed in order to determine clonality.

### 2.6. Whole-Genome Sequencing

Representative isolates from Clones 1 and 2 were cultured on blood agar plates, and total DNA was extracted from 10 colonies of each isolate using MagNA Pure Compact (Roche^®^, Mannheim, Germany). Libraries were prepared using the NEBNext^®^ Fast DNA Fragmentation & Library Prep Set for Ion Torrent™ (New England Biolabs, Ipswich MA, USA) according to the manufacturer’s instructions. The purification steps were performed using Mag-Bind^®^ TotalPure NGS beads (Omega Bio-Tek, Norcross, GA, USA). Libraries were sequenced using Ion Chef and S5 Gene Studio (ThermoFisher Scientic, Waltham, MA, USA) and assembled using the Nullarbor pipeline [13] and Geneious (v.10.1.3, Biomatter Ltd., Auckland, New Zealand). Plasmid Finder (https://cge.cbs.dtu.dk (accessed on 5 July 2021)) was used in order to search for plasmids in the sequenced genomes. The genomes were deposited in GenBank under the accession numbers JADKMB000000000, JADOZB000000000, JADOZC000000000, JADIXN000000000, JADOVL000000000, and JADOVK000000000.

### 2.7. Retrieving Clinical Microbiology Records

Clinical, microbiological, and epidemiological data were retrospectively retrieved from the hospital’s information system database. These data include clinical information, intestinal colonization status after identification through matrix-assisted laser desorption/ionization-time of flight (MALDI-TOF), antibiotic susceptibility testing (AST), and extraintestinal *Serratia* spp. isolates. AST was performed using the NC82 panels of the MicroScan Walk-Away 96 Plus (Beckman Coulter, Miami, FL, USA) and interpreted according to the EUCAST breakpoints [14]. MALDI-TOF was performed using single colonies grown overnight on MacConkey agar plates using the MBT-Smart Biotyper (Bruker Daltonics, Bremen, Germany) and the MBT Compass reference library (Version 4.1) according to the manufacturer’s instructions. Statistical tests were performed using SPSS (version 19.0, IBM, New York, NY, USA), and *p*-values < 0.05 were considered significant. Ethical approval was obtained from the local ethics committee (PI-3428).

## 3. Results

### 3.1. Patients, Outbreaks, and Samples

In late December 2018, *Serratia marcescens* was detected in rectal swabs obtained from five different neonates through routine epidemiological screening at the neonatal ward at the Hospital Universitario La Paz (HULP). These episodes occurred within the same week and well above the baseline of detecting, at most, one to two sporadic incidents of *Serratia* spp. per week. This led to the declaration of an outbreak by the Preventive Medicine Department. The outbreak extended over 13 weeks, where *S. marcescens* was detected in the rectal swabs from a total of 13 neonates through routine culture-based surveillance (Figure 1). These patients are hereafter referred to as Group 1 neonates. All the rectal swabs that were received from Group 1 neonates during this time period (including those that were negative through routine cultures) were stored. Rectal swabs from 29 other neonates that were present at the hospital’s neonatal ward but did not have any swab that was positive for *Serratia* spp. through culture were also collected (hereafter referred to as Group 2 neonates). Strict infection-control measures that include rigorous cleaning and disinfection of the ward, patient-contact precautions, and limiting visits, among others, were implemented. Thirteen weeks after the onset of the outbreak, it was declared to be over. After these 13 weeks, *Serratia* spp. was not detected in any rectal swab originating from the neonatal ward through routine culture for the following three weeks.

On average, each neonate involved in the outbreak was hospitalized for 5.3 weeks (range: 1–8 weeks). Table 1 shows the baseline clinical characteristics of the patients involved in the study, and Table 2 shows details regarding the main diagnoses and clinical outcomes. Though the main diagnoses varied greatly between the patients, seven patients from Group 1 and thirteen patients from Group 2 were premature. Eleven patients from Group 1 and thirty-six patients from Group 2 were discharged from the hospital during the study, while the rest of the patients passed away.

Fifty-two rectal swabs were collected from the Group 1 neonates (Figure 1) and 58 from Group 2 neonates (total swabs = 110). MALDI-TOF identified all the isolates obtained from Group 1 neonates as *S. marcescens*, with scores > 2.5. On average, four samples per patient were taken, and the average number of days between the first and last sample was 37. Regarding Group 2, multiple samples were taken from five neonates (range: 2–4 samples), with an average of 15 days between the first and last samples. Only one sample was obtained from the rest of the neonates in this group.

Eight out of the thirteen neonates of Group 1 had extraintestinal isolation of *Serratia* sp. (Table 3). Two of these patients (P6 and P18) have had more than one episode of detection of *Serratia* sp. from the same site (at least two weeks apart), while two patients (P3 and P18) have had *Serratia* spp. isolated from several body sites.

*Serratia* spp. caused infections in seven out of the eight patients where it was detected extraintestinally. These infections were sepsis (*n* = 2), conjunctivitis (*n* = 2), pneumonia (*n* = 1), both sepsis and meningitis (*n* = 1), and bacteraemia (*n* = 1). Notably, all eight patients with extraintestinal *Serratia* spp. isolates have had overlapping stays in the seven-bed neonatal intensive care unit throughout the outbreak. Regarding Group 2 neonates, there was only one episode of extraintestinal isolation of an unrelated *Serratia liquefaciens* from the blood of one of the patients.

AST profiles showed that all extraintestinal isolates were resistant to ampicillin, amoxicillin/clavulanic acid, cephalothin, cefuroxime, cefoxitin, and tobramycin and susceptible to the rest of the tested antibiotics. Through the NC82 panels of the MicroScan Walk-Away 96 Plus, all the isolates were urease-positive after 24 h.

### 3.2. Random Amplified Polymorphic DNA (RAPD) Analysis

Clonality analysis using RAPD was performed for the first isolate obtained from the neonates of Group 1, in addition to a bloodstream isolate from one of these patients and four environmental samples. RAPD analysis showed that seven patients had isolates pertaining to the same clone (Clone 1). All of them were isolated between weeks one and eight of the outbreak. The same clone was detected in the environmental samples obtained during week six (Figure 2 and Appendix A). The two other isolates collected from the two remaining neonates that were affected by the outbreak in the first eight weeks were clonally unrelated. Three out of the four isolates obtained from the neonates affected between weeks eight and thirteen belonged to a different clone (Clone 2). The fourth isolate obtained in this period was clonally unrelated.

### 3.3. Whole-Genome Sequencing

Whole-genome sequencing was performed for three isolates of Clone 1 (S1, S3B, and S9), including the blood isolate (S3B) and the three isolates that belonged to Clone 2 (S17, S18, and S19). Appendix A shows the parameters obtained from the de novo assembly of these isolates.

Pan-genome analysis using the Nullarbor pipeline detected 6565 genes and showed that isolates of Clone 1 clustered together, while those of Clone 2 belonged to another cluster (Figure 3). A total of 3286 genes were common between all six isolates (core genes). From the 3279 shell genes analysed (i.e., genes that are not common among all isolates), the genes of Clone 1 clustered together, while those of Clone 2 clustered together. The Clone 1 isolates had seven to twelve core-genome single-nucleotide polymorphisms (SNPs) between them, while those of Clone 2 had zero to seven SNPs. The number of core-genome SNPs, comparing the Clone 1 isolates to the Clone 2 isolates, was above 1000 SNPs.

Genbank’s Average Nucleotide Identity pipeline (https://www.ncbi.nlm.nih.gov/WebSub/ (accessed on 26 July 2021)) identified some isolates as *Serratia marcescens* and others as *Serratia ureilytica*. The sequences were then remapped against an *S. marcescens* genome (Genbank accession numbers: CP021164.1-CP021164.2) and an *S. ureilytica* genome (Genbank accession numbers: PGPC01000001-PGPC01000004). The mapping showed that strains S1, S3B, and S9 (collected from week one to week eight) clustered with the *S. marcescens* genome, while isolates S17, S18, and S19 clustered with the *S. ureilytica* genome. Table 4 shows the antibiotic-resistance genes detected. No plasmids were detected in any of these isolates.

### 3.4. Relative Quantification of the Intestinal Load of Serratia *spp.*

The relative intestinal loads (RLs) of *Serratia* spp. were determined by the ∆∆C_t_ method. *Serratia* spp. was detected by qPCR in 51 out of the 52 swabs obtained from Group 1 (there was one sample negative by qPCR and by culture) and in 19 out of the 58 swabs from Group 2 (originating from eleven patients). Fourteen samples that were positive by qPCR in Group 1 and all the samples from Group 2 were negative by culture. The *16SrRNA* gene was amplified in all the samples, and no significant difference was detected in its raw C_t_ values between all the samples collected. Figure 4 shows the RLs of the positive rectal swabs for *Serratia* spp. for all the patients included in our study and for which more than one sample was collected.

The average RL of *Serratia* spp. obtained from swabs of Group 1 neonates was 6.31% (Log∆∆C_t_ = −1.02 ± 0.7) and significantly higher than those of Group 2 (0.09%; Log∆∆C_t_ = −3.07 ± 1.5; *p* < 0.05; Figure 5A). The first swab collected from each of the Group 1 neonates had higher average RLs (26.9%; Log∆∆C_t_ = −0.57 ± 0.47) as compared to the last swabs obtained from these neonates (4.37%; Log∆∆C_t_ = −1.36 ± 0.8; *p* < 0.05; Figure 5B).

The average RL of *Serratia* spp. in patients with extraintestinal detection of this organism (in swabs collected within the same week) was higher than those that did not have extraintestinal spread: (2.78% (Log∆∆C_t_ = −1.55 ± 0.5) versus 0.29% (Log∆∆C_t_ = −2.53 ± 1.3); *p* < 0.05). Moreover, in the two patients where *Serratia* spp. was isolated from several body sites (P3 and P18), the RLs in the time frame when the isolates were being detected were much higher than the average (range: 55%–36.31%; (Log∆∆C_t_s = −0.26 to −0.44; Figure 3)).

## 4. Discussion

In this study, we investigated an outbreak caused by *Serratia* spp. that took place at the neonatal ward of the Hospital Universitario La Paz (Madrid, Spain) and extended for 13 weeks. *Serratia* spp. are environmental organisms that harbour several virulence factors, which allows them to be successful in causing outbreaks. One of these factors is the ability to produce biofilms, which makes them capable of attaching to different hospital surfaces and gives them a degree of protection against antimicrobial agents. Other virulence factors include the production of haemolysins and proteases that induce cytotoxicity, as well as their swarming ability, which allows them to migrate to places where they can develop and persist [15]. In addition to their virulence factors, *Serratia* spp. are intrinsically non-susceptible to several antibiotics that include second-generation cephalosporins and tetracyclines [4]. They have also demonstrated the ability to readily acquire genes of antibiotic resistance through horizontal gene transfer, converting them into organisms that are non-susceptible to a wide range of broad-spectrum antibiotics [16].

RAPD analysis and whole-genome sequencing showed that the first eight weeks of the outbreak was caused by *Serratia marcescens*, while *Serratia ureilytica* was responsible for the remaining five weeks. These two species are very closely related organisms that have 98.3% *16SrRNA* gene-sequence similarity. Moreover, though both have ureolytic activities, they are only phenotypically distinguishable by the higher rate of ureolytic activity for *S. ureilytica* [17]. The only typing method currently available that is able to distinguish between *Serratia* spp. and that is universally accepted is next-generation sequencing. However, recent attempts at developing schemes for the differentiation of different *S. marcescens* using three housekeeping genes have been developed [18]. This method might be useful to track different clones during an outbreak, but its ability to distinguish between different species of *Serratia* has not yet been tested.

To the best of our knowledge, this is the first report where *S. ureilytica* is implicated in a human outbreak. However, due to its similarity to *S. marcescens*, it is possible that this organism had already caused several outbreaks but was misidentified as another member of *Serratia* spp., as it was in our case. Further investigation of the reason for misidentification showed that the reference library in the MALDI TOF biotyper used for routine surveillance at HULP contains only one profile for *S. ureilytica*, which is very close to that of *S. marcescens*, making it possible to misidentify this former species (Appendix A).

Culture-based screening led to the identification of thirteen neonates that were affected by this outbreak. Nevertheless, through qPCR, eleven additional patients were found to be colonized by *Serratia* spp. We showed that the RLs of *Serratia* spp. were significantly higher among patients that were both culture- and qPCR-positive (Group 1) as opposed to culture-negative and qPCR-positive patients (from Group 2). There was a small overlap of RLs between these two patient populations, but the RLs that were below 1% (Log∆∆C_t_ = −2) among Group 1 patients were in samples collected towards the end of their involvement in the outbreak (Figure 4 and Figure 5) and mostly from samples that were culture-negative (Figure 1). Furthermore, the last samples collected from the outbreak patients had significantly lower RLs as compared to the first samples. Taken together, these data demonstrate that during the outbreak, the gut microbiome of these patients was dominated by *Serratia* spp. This is a phenomenon that was previously reported for other organisms [19,20] and linked to sepsis [21]. Notably, eight out of the thirteen Group 1 patients had extraintestinal *Serratia* spp. isolates, and one patient from Group 2 had *Serratia* spp. isolated from a blood sample. These patients had higher RLs at the time of extraintestinal detection of *Serratia* spp. as compared to patients in which this organism was not detected outside the intestines. Moreover, eight of these patients were actually infected by *Serratia* spp., and this bacterium was isolated from the blood in five independent episodes. This finding suggests a relationship between intestinal dominance by *Serratia* spp. and their ability to escape their niche, move to extraintestinal sites, and cause infections.

Horizontal transmission of *S. marcescens* among low-birthweight preterm infants has also been linked to intestinal dominance [22]. Our data, though limited in quantity, hint at this relation since *Serratia* spp. was able to colonize several extraintestinal sites, including water reservoirs. Though in this study, we only investigated extraintestinal spread within the same patient, being able to escape the intestine might be an important step in horizontal transmission. This, combined with the fact that the outbreak patients coincided in the same ward in overlapping time frames during the outbreak, reinforces the relationship between intestinal dominance and horizontal transmission of *Serratia* spp.

The approach that we used to determine the RLs of *Serratia* spp. allows for the determination of the extent of intestinal dominance. Though the small number of patients in our study does not allow for global conclusions, determining RLs over time could have the potential to predict the extraintestinal spread of these organisms since the patients with extraintestinal spread of *Serratia* spp. had higher RLs as compared to those that did not. From the data collected in our study, we propose using the qPCR-based approach to determine intestinal dominance in the setting of an outbreak due to the critical information that it provides. Further exploring the clinical usefulness of the early detection of intestinal dominance could, in turn, have very important implications in preventive medicine and infection control.

## Figures and Tables

**Figure 1 microorganisms-09-02271-f001:**
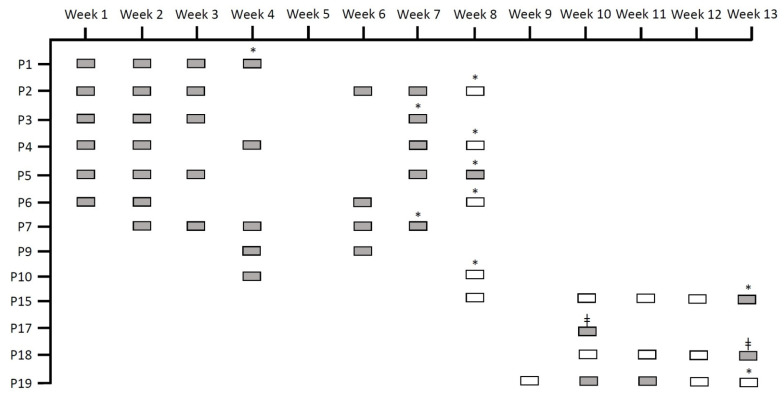
Rectal swabs obtained through routine epidemiological screening for *Serratia* spp. that were reused in our study. “P” stands for patient, grey rectangles indicate rectal swabs that were positive through routine culture, and white rectangles indicate rectal swabs that were negative through routine culture. All the rectangles depict samples that were included in our study. * Signifies that the patient was discharged, and ǂ signifies exitus.

**Figure 2 microorganisms-09-02271-f002:**
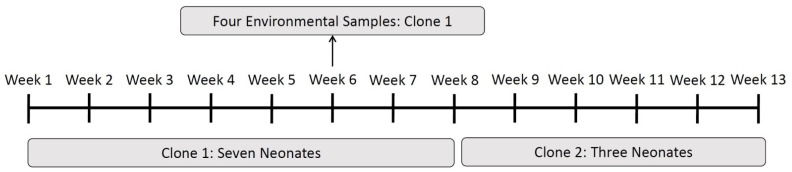
Distribution of the two clones detected by random amplified polymorphic DNA analysis over the 13 weeks of the outbreak.

**Figure 3 microorganisms-09-02271-f003:**
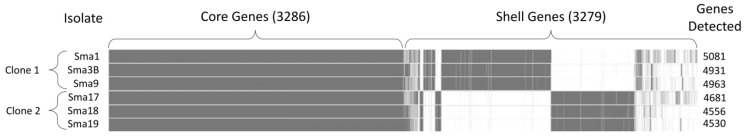
Pan-genome analysis for three isolates belonging to Clone 1 and three isolates belonging to Clone 2. The core genes are those common among all six isolates, while the shell genes are those that were different between the two clones.

**Figure 4 microorganisms-09-02271-f004:**
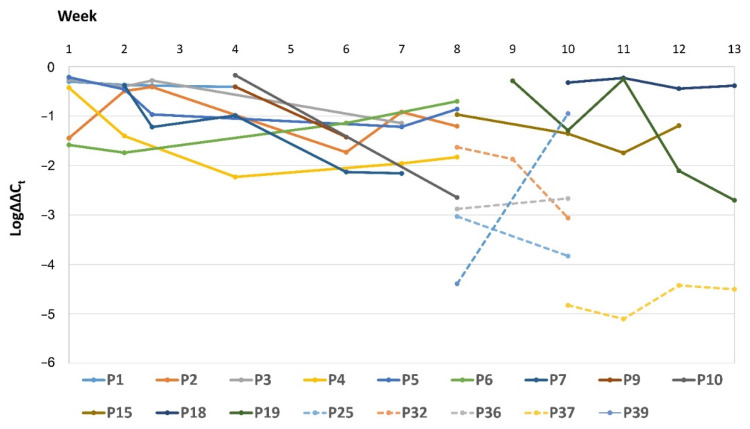
Relative intestinal loads of *Serratia* spp. over time for all the patients that have had more than one episode of detection of this organism in rectal swabs. The dotted line indicates the relative loads for patients of Group 2 (i.e., samples obtained from culture-negative patients). Log∆∆C_t_ values of 0 represent 100% of the bacterial population, −1 is 10%, −2 is 1%, and so on until −6 (0.0001%), which is the detection limit.

**Figure 5 microorganisms-09-02271-f005:**
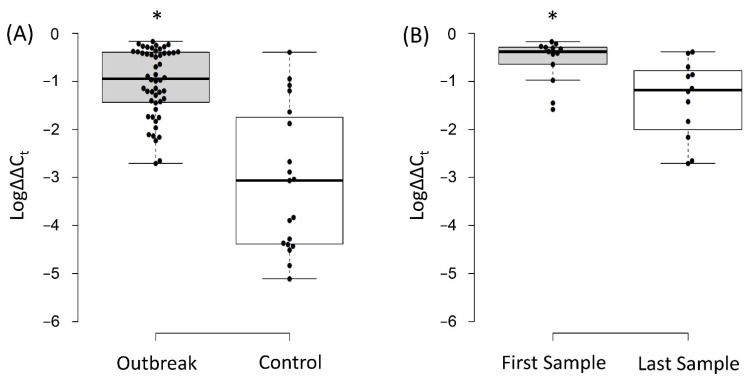
(**A**) Log∆∆C_t_ values of *Serratia* spp. in samples obtained from Group 1 neonates (that had positive cultures) compared to those obtained from Group 2 neonates (culture-negative neonates). (**B**) Log∆∆C_t_ values of *Serratia* spp. in the first swab collected from each neonate of Group 1 compared to the last swab collected. Log∆∆C_t_ values of 0 represent 100% of the bacterial population, −1 is 10%, −2 is 1%, and so on until −6 (0.0001%), which is the detection limit. The “*” indicates statistical significance (*p* < 0.05).

**Table 1 microorganisms-09-02271-t001:** Baseline clinical characteristics of the patients involved in this study. GA = gestational age; SD = standard deviation.

	Group 1 Neonates	Group 2 Neonates
Number of Patients	13	29
Days of Life Upon Inclusion (Mean ± SD)	17.5 ± 7.1	23.5 ± 36.9
GA, weeks (Mean ± SD)	32.2 ± 4.9	33.2 ± 5.1
Average Birth Weight (Grams)	1891.8 ± 935.5	2093.7 ± 1136.9
Female (%)	38.5%	37.9%
Premature (GA < 37 Weeks) (%)	76.9%	62.1%

**Table 2 microorganisms-09-02271-t002:** The main diagnosis and patient outcome of the patients involved in this study.

**Group 1 Neonates**		
**Patient**	**Main Diagnosis**	**Patient Outcome**
**P1**	Premature	Discharge
**P2**	Premature, patent ductus arteriosus (PDA) with surgery, posthemorrhagic ventriculomegaly	Discharge
**P3**	Premature, intestinal volvulus with surgery	Discharge
**P4**	Polymalformative syndrome	Discharge
**P5**	Premature	Discharge
**P6**	Premature, pulmonary hypertension (PH), stroke	Discharge
**P7**	Premature, bronchopulmonary dysplasia (BPD)	Discharge
**P9**	Congenital diaphragmatic hernia (CDH)	Discharge
**P10**	Epidermolysis	Discharge
**P15**	Congenital myopathy	Discharge
**P17**	Congenital cardiopathy, extracorporeal membrane oxygenation (ECMO)	Exitus
**P18**	Premature, intestinal perforation, chylothorax	Exitus
**P19**	Congenital atrioventricular block	Discharge
**Group 2 Neonates**		
**Patient**	**Main Diagnosis**	**Patient Outcome**
**P20**	Oesophageal atresia	Discharge
**P21**	Premature, PDA with surgery, PH	Discharge
**P22**	Premature, PDA with surgery, BPD, extensive leukomalacia	Discharge
**P23**	Congenital hyperinsulinism	Discharge
**P24**	Premature, Pulmonary Hypoplasia, BPD	Discharge
**P25**	Noonan Syndrome, hypertrophic cardiomyopathy	Exitus
**P26**	Congenital heart disease: Transposition of the Great Arteries (TGA)	Discharge
**P27**	Congenital chylothorax	Exitus
**P28**	Congenital heart disease: total pulmonary venous drainage, kidney failure	Exitus
**P29**	Hyaline membrane disease, pneumothorax	Discharge
**P30**	CDH, PH	Discharge
**P31**	Premature, BPD	Discharge
**P32**	Premature, PH, posthemorrhagic ventriculomegalia	Discharge
**P33**	Premature	Discharge
**P34**	CDH, PH	Exitus
**P35**	Premature, BPD	Discharge
**P36**	Oesophageal atresia	Discharge
**P37**	Premature	Discharge
**P38**	Premature, congenital tuberculosis	Discharge
**P39**	Congenital hyperinsulinism	Discharge
**P40**	Premature, BPD	Discharge
**P41**	Transfer at 3 months of life, premature, BPD, necrotizing enterocolitis (NEC)	Discharge
**P42**	Premature	Discharge
**P43**	Premature	Discharge
**P44**	Congenital heart disease: Tetralogy of Fallot	Discharge
**P45**	Congenital heart disease: total pulmonary venous drainage, kidney failure	Discharge
**P46**	Hirschsprung disease, intestinal perforation, ileostomy	Discharge
**P47**	Premature	Discharge
**P48**	Oesophageal atresia	Discharge

**Table 3 microorganisms-09-02271-t003:** Extraintestinal detection of *Serratia* spp. among the Group 1 patients. GA = gestational age; ND = not determined; Age (days) = age upon inclusion in the study.

Patient	Gender	GA (Weeks)	Age (Days)	Prior Colonization	Previous Antibiotics	Extraintestinal *Serratia* spp.
**P1**	Female	32.1	13	No	No	Conjuctiva (*conjunctivitis*)
**P2**	Male	26.5	27	No	Yes	Bronchial aspirate (*pneumonia*)
**P3**	Male	32.6	14	Yes	Yes	Blood, cerebrospinal fluid, skin wound, and skin around ileostomy (*sepsis* and *meningitis*)
**P4**	Male	37.6	11	ND	Yes	None
**P5**	Male	26	27	Yes	Yes	Blood (*bacteriemia*)
**P6**	Male	27.5	24	No	Yes	Conjuctiva (two isolates) (*conjunctivitis*)
**P7**	Female	29.4	18	ND	Yes	None
**P9**	Female	37.2	14	ND	Yes	None
**P10**	Female	35	15	ND	Yes	None
**P15**	Male	36.5	29	ND	No	None
**P17**	Male	38	11	Yes	Yes	Bronchial aspirate (*colonization*)
**P18**	Female	24.4	17	Yes	Yes	Blood (two isolates), bronchial aspirate, pericatheter skin, and skin around ileostomy (*sepsis*)
**P19**	Male	35.2	7	No	Yes	Blood (*sepsis*)

**Table 4 microorganisms-09-02271-t004:** Genes of resistance detected in the six isolates that were chosen for whole-genome sequencing.

Strain	Gene (Antibiotic Resistance)					
	*qnrB56* (Quinolones)	*aac(6′)-Ic* (Aminoglycosides)	*bla*_SRT-2_ (β-Lactams)	*bla*_SST-1_ (β-Lactams)	*oqxB* (Efflux Pump)	*tetA41* (Tetracycline)
**S1**	✕	✓	✕	✓	✓	✓
**S3B**	✕	✓	✕	✓	✓	✓
**S9**	✕	✓	✕	✓	✓	✓
**S17**	✓	✓	✓	✕	✓	✕
**S18**	✓	✓	✓	✕	✓	✓
**S19**	✓	✓	✓	✕	✓	✓

## Data Availability

All data generated or analysed during this study are included in this article and its Appendix A. Further enquiries can be directed to the corresponding author (ED).

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
