# Peer review of "Intestinal Dominance by Serratia marcescens and Serratia ureilytica among Neonates in the Setting of an Outbreak"

_microorganisms, 2021, doi:10.3390/microorganisms9112271_

Round 1

Reviewer 1 Report

General:

please adhere to the International guidelines for writing bacterial names (first mention, subsequent mention, in italics…)

please discuss the intrinsic non-susceptibilities of Serratia species. Either use the EUCAST Expert Rules and intrinsic resistance or refer to the summary table form the following article as an example:

https://doi.org/10.3390/life11101059

please define all abbreviations on first mention

in many places, commas, Oxford commas are missing.

Abstract:

please format the abstract according to the Instructions for authors

what do Groups 1 and 2 represent? this is unclear from the abstract, please clarify or remove

L28: why is it so capable of causing outbreaks? please elaborate on this further!

may be challenging

methods:

2.1. section should be split into two different sections

L86-90: please try to explain this more clearly, as the current form is very confusing!

please provide more technical details associated with the MALDI measurements!

The quality of Figure 2. should be improved.

The quality of Figure 3. should be improved, the legend cannot be read in its current form.

L283: extra dot

In the discussion, please disucss presently available and future typing methods, that may be relevant in the following and control of nosocomial outbreaks of Gram-negative bacteria.

Please discuss more the virulence and the AMR of Serratia, and provide more references in this context.

Author Response

Dear Reviewer,

I would like to first thank you for taking the time for revising our manuscript and for providing us with a constructive and informative review. Please find below our point-by-point reply to your comments:

General:

please adhere to the International guidelines for writing bacterial names (first mention, subsequent mention, in italics…)

Reply: The manuscript was reviewed and the bacterial names were modified according to the International Guidelines for Writing Bacterial Names.

please discuss the intrinsic non-susceptibilities of Serratia species. Either use the EUCAST Expert Rules and intrinsic resistance or refer to the summary table form the following article as an example: https://doi.org/10.3390/life11101059

Reply: The intrinsic non-susceptibilities of Serratia species have been expanded upon and the EUCAST Expert Rules have been sited (lines 36-41 and 307-310).

please define all abbreviations on first mention

Reply: The manuscript was reviewed and the abbreviations defined on first mention.

in many places, commas, Oxford commas are missing.

Reply: The manuscript was reviewed and the appropriate commas were added.

Abstract:

please format the abstract according to the Instructions for authors

Reply: The abstract was re-formatted according to the instructions for authors.

what do Groups 1 and 2 represent? this is unclear from the abstract, please clarify or remove

Reply: The sentences introducing Group 1 and Group 2 neonates were re-written in order to clarify that Group 1 neonates are those that have had Serratia spp. isolated through culture media and Group 2 neonates are those that did not have Serratia spp. isolated through culture media (lines 12-15).

L28: why is it so capable of causing outbreaks? please elaborate on this further!

Reply: Serratia spp.’s ability to cause outbreaks was elaborated upon (lines 31-33 and 301-307).

may be challenging

Reply: “can” was changed to “may” in line 37.

methods:

2.1. section should be split into two different sections

Reply: This section was split in two different sections as follows:

2.1. Sample Collection from Neonates During the Thirteen Weeks of the Outbreak

2.2. Water Samples Collection

L86-90: please try to explain this more clearly, as the current form is very confusing!

Reply: The explanation of the expression of the results was re-written in order to add clarity as such (lines 104-107):

“The results were expressed as Relative intestinal Loads (RLs) in an inverse logarithmic scale where a value of 0 represents 100% of the bacterial population, -1 represents 10% of the bacterial population, -2 represents 1% of the bacterial population, and so on until -6 (our detection limit) that represents 0.0001% of the bacterial population.”

please provide more technical details associated with the MALDI measurements!

Reply: Details regarding the MALDI-TOF technique and the cutoff scores for species identification (>2.5) were added in sections 2.7 (line 150-151) and 3.1 (line 196), respectively.

The quality of Figure 2. should be improved.

Reply: A better quality figure was produced (now called Figure 3).

The quality of Figure 3. should be improved, the legend cannot be read in its current form.

Reply: A better quality figure was produced (now called Figure 4).

L283: extra dot

Reply: The extra dot was removed.

In the discussion, please disucss presently available and future typing methods, that may be relevant in the following and control of nosocomial outbreaks of Gram-negative bacteria.

Reply: The typing methods currently available that could be used in the setting of Serratia spp. outbreaks have been mentioned (lines 317-323). However, we feel that discussing the typing methods available for all Gram-negative bacteria falls beyond the scope of this article.

Please discuss more the virulence and the AMR of Serratia, and provide more references in this context.

Reply: The virulence and the AMR of Serratia was discussed, with more references provided (lines 32-33, 36-41, and 301-311).

Reviewer 2 Report

The subject of the manuscript is quite interesting, but in its present form, due to the chaotic nature of the presentation, it requires a thorough re-editing.

Materials and methods:

Line 61: where were the water samples collected from? The results are not discussed later in the manuscript

Susceptibility assessment was not included in this section, but susceptibility profiles are described in the results.

The part about the number, method and source of the trials is unclear. Unfortunately, the diagram in Fig 1 does not help to understand. Perhaps this is also why the results are described in a rather vague manner.

What were the criteria for selecting strains for sequencing?

An electrophoregram with RAPD results would be necessary to validate the results

Author Response

Dear Reviewer,

I would like to first thank you for taking the time for revising our manuscript and for providing us with a constructive and informative review. Please find below our point-by-point reply to your comments:

Materials and methods:

Line 61: where were the water samples collected from? The results are not discussed later in the manuscript

Reply: The source and methods for the collection of the water samples were presented in section 2.1 in the original version to the manuscript, but are now moved to section 2.2 in order to add clarity. They are discussed later in the manuscript in lines: 124 (included in RAPD analysis), 224-225 (being the same clone as Clone 1), 356 (previously it was written at fomites and now it is changed to water reservoirs), and the newly added Figure 2 and Supplementary Figure S2.

Susceptibility assessment was not included in this section, but susceptibility profiles are described in the results.

Reply: As was mentioned in section 2.6 of the original manuscript (now section 2.7; lines 144-155), the antibiotic susceptibility profiles were retrieved from the hospital’s information system database, and not performed as part of this study.

The part about the number, method and source of the trials is unclear. Unfortunately, the diagram in Fig 1 does not help to understand. Perhaps this is also why the results are described in a rather vague manner.

Reply: The authors are not sure they understand the question correctly. In any case, we believe that we needed to highlight the fact that this study was retrospective in nature, and did not happen in real time as the outbreak was ongoing. This was highlighted in lines 47-48 and 53-55.

As for presenting the results in a vague manner, we did extensive revisions to the manuscript in order to add clarity. If the reviewer thinks that the results are still presented vaguely, then the authors would appreciate some examples and will improve on the clarity of presenting the results.

What were the criteria for selecting strains for sequencing?

Reply: The criteria for selecting strains for sequencing were choosing isolated that belonged to Clone 1 and isolates that belong to Clone 2 as determined by RAPD in order to compare their genomes. This was mentioned in the manuscript in lines 130 and 235-237.

An electrophoregram with RAPD results would be necessary to validate the results

Reply: A representative electrophoregram of this analysis was added as a supplementary figure (S2).

Reviewer 3 Report

This manuscript by Elias Dahdouh et al. describe interesting data on a outbreak due to Serratia among neonates.

This manuscript deserves revision before possible acceptance for publication.

Global :

Prefer passive form

Italicize "i.e."

numbers below 12 have to be written in full letters.

Methods : 

Manufacturer's information have to be indicated once but indicating name/city/country and not country alone.

Why have the authors decided to sample on this frequency and on this period of time? Justification are not sufficient even if justified by "routinely collected for the epidemiological surveillance".

Results : 

Data included in suppl. table 1 have to be included in the main core manuscript.

A figure, summarizing the RAPD analysis results could be of interest to add to the manuscript.

Discussion : 

Did the authors conclude, at the end of their study on two different outbreaks? How did they explain the introduction of S. ureilytica?

Author Response

Dear Reviewer,

I would like to first thank you for taking the time for revising our manuscript and for providing us with a constructive and informative review. Please find below our point-by-point reply to your comments:

Global :

Prefer passive form

Reply: The manuscript was revised and the relevant sentences were modified in order to use the passive form.

Italicize "i.e."

Reply:i.e.” was italicized throughout the manuscript

numbers below 12 have to be written in full letters.

Reply: The manuscript was revised and numbers below 12 were written in full letters.

Methods :

Manufacturer's information have to be indicated once but indicating name/city/country and not country alone.

Reply: The manufacturer’s information has been updated to include the city.

Why have the authors decided to sample on this frequency and on this period of time? Justification are not sufficient even if justified by "routinely collected for the epidemiological surveillance".

Reply: The sampling frequency paralleled that of the epidemiological surveillance due to the retrospective nature of the study. Clarifications in the manuscript were added to highlight that the study was retrospective (lines 47-48 and 53-55).

Results :

Data included in suppl. table 1 have to be included in the main core manuscript.

Reply: Supplementary table one is now included in the main core manuscript (Table 2), in addition to summarizing some of the data included in the table in lines 185-188.

A figure, summarizing the RAPD analysis results could be of interest to add to the manuscript.

Reply: A figure summarizing the RAPD analysis results was added (Figure 2), and a representative image showing the different profiles obtained was added as Supplementary Figure S2.

Discussion :

Did the authors conclude, at the end of their study on two different outbreaks? How did they explain the introduction of S. ureilytica?

Reply: The outbreak was defined as a single outbreak from an epidemiological perspective and as it was perceived by the Preventive Medicine Department at the time of the outbreak. The retrospective analysis of this outbreak demonstrated the presence of two organisms. However, the initial assumption was not changed because this information was received a posteriori.

As for the introduction of S. ureilytica, we believe that it was introduced in the same was as S. marcescens since they are both environmental opportunistic pathogens whose reservoir is not easily determined. While at week six we received water samples where we were able to detect S. marcescens, no further water samples were received in order to determine whether S. ureilytica could also be isolated from that source. Therefore, we cannot conclude on how S. ureilytica was introduced, but we could speculate that it was introduced from an environmental source. If the reviewer sees that it is necessary to include all or part of this information in the manuscript, then the authors will gladly do so.

Round 2

Reviewer 1 Report

The main concerns were addressed during the revisions.

Reviewer 2 Report

The authors responded to all the uncertainties

I have no more objections